# Melanoma Recognition by Fusing Convolutional Blocks and Dynamic Routing between Capsules

**DOI:** 10.3390/cancers13194974

**Published:** 2021-10-03

**Authors:** Eduardo Pérez, Sebastián Ventura

**Affiliations:** 1Knowledge Discovery and Intelligent Systems in Biomedicine Laboratory Maimónides Biomedical Research Institute of Córdoba, 14004 Córdoba, Spain; eduardo.perez@imibic.org; 2Department of Computer Science and Numerical Analysis, University of Córdoba, 14071 Córdoba, Spain; 3Department of Information Systems, King Abdulaziz University, Jeddah 21413, Saudi Arabia

**Keywords:** melanoma diagnosis, CapsNet, convolutional neural network, interpretation tool

## Abstract

**Simple Summary:**

The early treatment of skin cancer can effectively reduce mortality rates. Recently, automatic melanoma diagnosis from skin images has gained attention, which was mainly encouraged by the well-known challenge developed by the International Skin Imaging Collaboration project. The majority of contestant submitted Convolutional Neural Network based solutions. However, this type of model presents disadvantages. As a consequence, Dynamic Routing between Capsules has been proposed to overcome such limitations. The aim of our proposal was to assess the advantages of combining both architectures. An extensive experimental study showed the proposal significantly outperformed state-of-the-art models, achieving 166% higher predictive performance compared to ResNet in non-dermoscopic images. In addition, the pixels activated during prediction were shown, which allows to assess the rationale to give such a conclusion. Finally, more research should be conducted in order to demonstrate the potential of this neural network architecture in other areas.

**Abstract:**

Skin cancer is one of the most common types of cancers in the world, with melanoma being the most lethal form. Automatic melanoma diagnosis from skin images has recently gained attention within the machine learning community, due to the complexity involved. In the past few years, convolutional neural network models have been commonly used to approach this issue. This type of model, however, presents disadvantages that sometimes hamper its application in real-world situations, e.g., the construction of transformation-invariant models and their inability to consider spatial hierarchies between entities within an image. Recently, Dynamic Routing between Capsules architecture (CapsNet) has been proposed to overcome such limitations. This work is aimed at proposing a new architecture which combines convolutional blocks with a customized CapsNet architecture, allowing for the extraction of richer abstract features. This architecture uses high-quality 299×299×3 skin lesion images, and a hyper-tuning of the main parameters is performed in order to ensure effective learning under limited training data. An extensive experimental study on eleven image datasets was conducted where the proposal significantly outperformed several state-of-the-art models. Finally, predictions made by the model were validated through the application of two modern model-agnostic interpretation tools.

## 1. Introduction

Cutaneous malignant melanoma is on the rise and has the highest mortality rate among the various types of skin cancer [1]. For example, in 2021, it is estimated that 106,110 new cases of melanoma will be diagnosed in the United States, resulting in 7180 deaths (https://www.cancer.org, accessed on 1 June 2021). Surgery is the primary treatment for this type of cancer, but in its more advanced stages, treatment can also include immunotherapy, targeted therapy drugs and radiation to extend survival. Accordingly, the development of modern tools is critical for diagnosing melanoma at an earlier stage, thus easing the decision-making process for dermatologists and reducing invasive treatments for patients, in addition to associated costs. The diagnosis of melanoma is, however, a complex task even for expert dermatologists, mainly because of the complexity, variability and ambiguity of symptoms [2]. Additionally, an extensive variety of morphologies exist even between samples from the same category, which greatly hampers diagnosis. Several studies have shown that the early diagnosis of melanoma can greatly benefit from computational methods [3], demonstrating that such techniques may even outperform dermatologists in terms of diagnosis [4], due to various machine learning techniques and learning of data-driven features for specific tasks [5]. The early proposed methods required the previous extraction of handcrafted features, thus relying on the level of dermatologists’ expertise to extract high quality descriptors. This extraction process of informative and discriminative sets of high-level features, however, remains as a complex and costly task that is usually problem dependent [6], and it is noteworthy that sometime is impossible to derive invariant features which are independent of the differences in the input images [7]. On the other hand, there is another type of computational method which can automatically extract and learn high-level features [8], providing a higher robustness to the inter- and intra-class variability present in melanoma images [8,9].

Deep learning models, specifically Convolutional Neural Network (CNN) models, have the capacity of automatically learning high-level features from raw images [8,10,11]. The ImageNet Challenge (ILSVRC) takes place every year since 2010. In 2012 a CNN won the contest for the first time, which increased the popularity of such models for image processing [12]. CNN models learn automatically abstract features and enable the learning for several tasks. For example, Pérez et al. [13] summarized the most popular techniques used in CNN models for diagnosing skin images. Furthermore, this type of deep model can extract sets of patterns ranging from single edges and curves to more complex patterns such as a human face. On the other hand, the main downside of CNN models is that the information regarding spatial relationships between extracted features is lost. For example, CNN models could consider two images to be similar if they share the same objects, even if the location within the image is relevant. However, convolution operation is not translation-invariant.

To overcome the above main limitation of CNN models, a new type of deep learning model, named Dynamic Routing Between Capsules (well-know as CapsNet), was proposed in [14], where the authors designed a method closer to how human vision works. The neurons in this architecture can represent properties of a object such as position, size and texture. Moreover, CapsNet is able to preserve hierarchical spatial relationships, and in theory it could be as effective as any CNN but using fewer samples for training [14]. Niyaz et al. [15] reviewed several deep learning methods for the prediction of different types of cancer. In that time the authors did not find evidence of the application of CapsNet in cancer diagnosis. However, the authors acknowledged CapsNet as a promising model for diagnosing cancer and encouraged its application. Accordingly, CapsNet has been applied in medical image analysis, demonstrating to be really effective for lung cancer screening [16], blood cell image classification [17], and cervical image classification [18], to list a few applications. Finally, CapsNet have been recently applied in skin cancer classification. Cruz et al. [19] used CapsNet to classify skin lesions using images and evaluated their proposal in only one recognized dataset, HAM10000 [20]. However, to our understanding, the proposal has several issues. Firstly, although skin images are usually high-quality (600×450×3 in HAM10000), the authors resized images to 28×28×1, losing a considerable amount of pixels and even ignoring colors in the images, which is important for the diagnosis of melanoma [21]. Secondly, the authors highlighted their performance relying mainly on overall precision. However, it is well-known that skin images datasets are unbalanced. Looking closely, the authors achieved only a precision of 28% and 41% in melanoma and basal cell carcinoma categories, respectively, leaving open a big margin of improvement.

Consequently, this work focuses on assessing the effectiveness of a new architecture for the diagnosis of melanoma. The architecture uses high-quality 299×299×3 skin lesion images and achieves an acceptable performance in both normal and malignant categories. The proposed architecture combines features from convolutional blocks and CapsNet. First, we selected a more sophisticated convolutional computational block, allowing for the extraction of more useful initial features. Second, we replaced the first convolutional block from CapsNet with the above computational block. As a result, we are able to extract more significant features from earlier stages. Next, primary caps extract geometric and color properties present in the images, such as asymmetry, border irregularity, color variegation and the positions of various zones. These have all proven to be very useful attributes to consider when diagnosing melanoma [21]. In this manner, we can maintain the hierarchical spatial relationships of patterns which yields great benefit. To take full advantage of the architecture, we proposes a hyper-tuning of the main parameters to ensure effective training and learning under limited training data. In addition, the architecture applies data augmentation to enhance the diagnosis of melanoma, significantly increasing the validity of the proposal. The new architecture enables the construction of a transformation-invariant model and the detection of spatial hierarchies between entities within an image. As such, it is more suitable for solving certain real-world situations than convolutional models. To evaluate the suitability of the proposal, an extensive experimental study was conducted on eleven public skin image datasets, allowing for a better analysis of the model’s effectiveness. The results showed that the proposed approach achieved very promising results and was competitive with respect to state-of-the-art CNN models which have previously been used in the diagnosis of melanoma. Finally, Shapley Additive Explanations method (SHAP (https://github.com/slundberg/shap, accessed on 1 September 2019)) [22] and Local Interpretable Model-agnostic Explanations (LIME) [23] were used to show the most important features and give a prediction with a high confidence level. This work, to the best of our knowledge, is the first attempt to thoroughly assess a new architecture based on convolutional blocks and CapsNet for the automatic recognition of melanoma. The hyper-parameters were specifically tuned for the selected task, achieving significantly better performance compared to the state-of-the-art models.

The rest of this work is arranged as follows: Section 2 briefly presents the state-of-the-art in solving the melanoma diagnosis problem mainly by using CNN models; Section 3 presents the proposed architecture; Section 4 presents the experimental study carried out, showing the results and a discussion of them; finally, some concluding remarks are presented in Section 5.

## 2. Related Works

CNN models have proven to be a powerful classification method for melanoma diagnosis [8]. This type of models presents a higher suitability compared to classic methods which depend on hand-crafted features. In addition, sophisticated techniques can be applied to even improve the performance of CNN models in the task of melanoma diagnosis, e.g., by applying data augmentation [24] and transfer learning techniques [8].

Data augmentation is a common technique applied to reduce overfitting on CNN models [25]. It is commonly performed by means of applying random transformations on the source images [26]. In addition, this technique can be used to tackle imbalance problems [27,28]. For example, Hossain and Muhammad [29] proposed an emotion recognition system using a CNN approach from emotional Big Data. The models trained with augmented data obtained better performance compared to its non-use. In addition, Esteva et al. [8] applied extensive data augmentation techniques during training; the authors increased the number of images by a factor of 720. Each image was randomly rotated, flipped and cropped. The results achieved a performance comparable to a committee of 21 dermatologists. On the other hand, more advanced techniques such as GANs are being applied to augment data [30]. GANs can augment a dataset by training simultaneously two models, a generator that creates new samples by randomly selecting points from the latent space, and a discriminator that determines whether a sample is a fake or not. Frid-Adar et al. [31] proposed methods based on GANs for generating synthetic medical images; their proposal was evaluated on a limited dataset of high quality liver lesion computed tomography. The results showed that the model increased both sensitivity and specificity by using augmented data.

Transfer learning is a technique widely used to increase performance when the number of training examples is limited [32,33]. This method transfers and reuses knowledge that was learned from a source task, where a lot of data is commonly available, e.g., the ImageNet dataset with more than one million of images. For instance, Esteva et al. [8] transferred the knowledge learned by InceptionV3 on ImageNet and applied it to melanoma diagnosis. Moreover, Nasr-Esfahani et al. [34] applied a pre-trained CNN to distinguishes between melanoma and nevus cases. The results showed that the proposed method is superior in terms of diagnostic accuracy in comparison with the state-of-the-art methods. Finally, Saba et al. [35] proposed an automated approach for skin lesion detection and recognition using Laplacian filtering, lesion boundary extraction and CNN. The results outperformed several existing methods and attained a high accuracy value.

On the other hand, CapsNet represents a completely novel type of deep learning architectures which attempt to overcome the limits and drawbacks of CNN models. Since CapsNet was recently proposed, only a few studies have explored its applications. Zhang et al. [18] applied CapsNet to classify the images of cervical lesions. The results showed better performance compared to other classification methods. Mobiny et al. [16] proposed an improvement on CapsNet that speedup the results compared to the original architecture. After evaluating the performance on computed tomography chest scans, the results showed that CapsNet is a promising alternative to CNN. Zhang et al. [36] combined CapsNet and fully CNN models in image scene classification, such as VGG16 and InceptionV3. The authors achieved better output compared to state-of-the-art methods. However, it is said that the use of a full CNN model could hamper the main aim behind CapsNet, which is the extraction of spatial hierarchies between entities. In addition, the number of trainable parameters significantly increases by combining such architectures. By demonstrating the benefits of CapsNet in medical imaging in this work, we may be encouraging its wider use. Considering the above, it would be interesting to design a deep learning architecture that combines and leverages features from different approaches such as data augmentation, transfer learning, convolutional blocks and CapsNet. After analyzing CapsNet, we strongly believe that specific blocks could be improved while maintaining their behavior. To augment data, it is important to perform a data augmentation both on training and test phases [24]. Next, the proposal for melanoma diagnosis, which follows the mentioned approximation, is described.

## 3. Materials and Methods

This section firstly describes the related works regarding the automatic diagnosis of melanoma from image data and the well-known state-of-the-art techniques, and then it presents the proposed architecture, which also uses the most proven techniques to date.

### 3.1. Proposed Architecture for the Diagnosis of Melanoma

Invariance and equivariance are two important concepts in image recognition area. To make a CNN transformation-invariant, a data augmentation of training samples is commonly performed. However, equivariance is a more general concept (invariance is a special case of equivariance) that allows a model detect the rotation or proportion change and adapt itself in a way that the spatial positioning inside an image is not lost [14]. This last requirement motivated the apparition of CapsNet networks.

CapsNet introduced the concept of capsule, where a capsule is a group of neurons or nested set of neural layers, and the state of the neurons inside a capsule can capture the properties of one entity inside an image. A capsule outputs a vector representing the instantiation parameters of a specific type of entity such as an object or a part of a object. In the other words, the output vector represents the probability of existence. Consequently, similar to the human vision process, these capsules are specialized at handling different types of stimulus and encoding things such as position, size, orientation, deformation, hue, texture, and other spatial information. The output vector can be calculated as
(1)$$v_{j}=\frac{\left|\right|s_{j}{\left|\right|}^{2}}{1+\left|\right|s_{j}{\left|\right|}^{2}}\frac{s_{j}}{\left|\right|s_{j}\left|\right|},$$
where $v_{j}$ and $s_{j}$ are the vector output of capsule *j* and its total output, respectively. The input to a capsule $s_{j}$ is a weighted sum over the vectors ${\hat{u}}_{j|i}$ in the layer below and is obtained as
(2)$$s_{j}=\sum_{i}c_{ij}{\hat{u}}_{j|i},\;\hat{u}=W_{ij}u_{i}$$
where $W_{ij}$ is a weight matrix and $c_{ij}$ are coefficients between capsule *i* and the rest of capsules in the layer above. The coefficients can be calculated as
(3)$$c_{ij}=\frac{exp(b_{ij})}{\sum_{k}exp\left(b_{ik}\right)},$$
where $b_{ij}$ are the log probabilities that capsule *i* should engage capsule *j*. Finally, CapsNet uses a separate margin loss, which can be calculated as
(4)$$L_{k}=T_{k}max(0,m^{+}-\left|\right|v_{k}{\left||\right)}^{2}+\lambda \left(1-T_{k}\right)max(0,||v_{k}\left|\right|-m^{-}{)}^{2},$$
where $T_{k}=1$ iff a sample of class *k* is present and $m^{+}=0.9$ and $m^{-}=0.1$. The total loss is the sum of the losses of nevus and melanoma capsules.

The baseline CapsNet architecture is composed by a simplistic Conv2D (256 filters, kernel 9×9, stride 1, ReLU activation function [37]), located at the beginning of the network, for extracting primary features which are subsequently passed to Primary and Class Caps layers. However, we hypothesized that CapsNet would attain a better performance if the first convolutional layer is replaced by a more sophisticated convolution-based computational block that was able to extract higher-level features before passing them to capsule layers. By this way, we leverage the benefits from both CNN and CapsNet for a better melanoma diagnosis.

Figure 1 shows the proposed architecture for the diagnosis of melanoma dubbed as MEL-CAP. The proposal was composed as follows: Input (299×299×3) → Customized convolutional block → Primary Caps (9×9, channels 32, capsule 16D) → Class Caps (2 capsules 64D, routing iteration 1). In addition, Table 1 shows a detailed description of every layer. After several phases of an experimental study, the above configuration was the most suitable for diagnosing melanoma. First, Inception architecture was considered given the effectiveness already demonstrated in the diagnosis of melanoma. Inception relies in independent convolutional blocks with filters that are powered the same input, which enables the extraction of more information over the same space. This architecture has been improved through the years, from V1 to V4 [38,39,40]. The latest updates showed that high performance could be also achieved by using aggressive dimension reductions, which allows to keep low hardware requirements.

By this way, we aimed a balance between the computational cost and the extraction of more high-level features before passing them to the capsule layers. Accordingly, Figure 1 also shows the convolutional block that replaced the first convolutional layer of CapsNet. The first block was composed as follows: Conv2D (32 filters, kernel 3×3, stride 2) → Conv2D (32 filters, kernel 3×3) → Conv2D (64 filters, kernel 3×3) → MaxPool2D (3×3, stride 2) → Conv2D (80 filters, kernel 1×1) → Conv2D (192 filters, kernel 3×3) → MaxPool2D (3×3, stride 2). The use of a convolutional block will not only allow more reduction of the input space, but also focusing on more important features from early stages. On the other hand, capsule layers, will not only learn richer patterns, but also paying more attention on learning their corresponding properties, such as location and orientation.

These abstract features learned by the first block are then passed as input to a convolutional capsule layer, named as primary caps, which is composed by 32 channels of convolutional 8D capsules with a 9 × 9 kernel and stride 2; i.e., in this case, each primary capsule comprises 8 convolutional units. These 8D capsules can identify features such as position, size, orientation, deformation, etc. The last layer, named as class caps, has two capsules 16D that represent the classes (nevus or melanoma), and these capsules receive input from all the capsules in the layer below. Moreover, as proposed in Sabour et al. [14], CapsNet uses a decoder block that influences the learning process, where this decoder intends to reconstruct an original image from the Class Caps layer representation. Finally, it is worth noting that CapsNet implements the routing mechanism mentioned earlier between two consecutive capsule layers (in our example between the layers primary caps and class caps), and this dynamic process can be viewed as a parallel attention mechanism that allows each capsule to attend to some active capsules at the level below and to ignore others [14].

### 3.2. Datasets

Table 2 shows a summary of the characteristics of the eleven datasets considered in this study, where all the images are labeled as nevus or melanoma. All the datasets were downloaded from The International Skin Imaging Collaboration (https://www.isic-archive.com, accessed on 1 September 2019) (ISIC) repository, except PH2 (https://bit.ly/39YEdmN, accessed on 1 September 2019) and MED-NODE (https://bit.ly/3DkCMvN, accessed on 1 September 2019) datasets. The MED-NODE dataset contains low resolution non-dermoscopic images taken with mobile phones. Nowadays, technological devices enables the collection of an enormous amount of data, which is essential for training models. On the other hand, PH2 dataset comprises high-quality dermoscopic images, where manual segmentation, clinical diagnosis and the identification of several dermoscopic structures were performed by expert dermatologists. The rest of datasets share the common characteristics of dermoscopic images. HAM10000 is the largest dataset in this work, which has been widely used in skin cancer diagnosis, e.g., Miglani and Bhatia [41] achieved 0.95 averaged AUC values for the overall classification. It can be observed that some datasets present a moderate imbalance ratio (ImbR), indicating that the number of nevus samples is several orders of magnitude higher than the number of melanoma samples, and this feature can commonly hamper the learning process of the machine learning models, e.g., MSK-3 and HAM10000. All the 16,601 images were resized to a resolution of h=299, w=299, and c=3, where *h* is the height, *w* is the width, and *c* is the number of channels of an image.

Table 2 also shows other insights about the data. For example, intra-class, inter-class distances and their ratio (DistR) indicate an important degree of similarity between categories. In addition, the silhouette score [42] indicated how much an image shares the same characteristics of its class compared to other classes. The above corroborated that even images from different classes are similar. Finally, in next Section the proposed architecture is evaluated and compared to state-of-the-art CNN models.

**Table 2 cancers-13-04974-t002:** Skin image datasets used in the experimental study; “ImbR”, “IntraC”, “InterC” and “DistR” represent the imbalance ratio between the normal and melanoma classes, the average distance between images of the same category, the average distance between images of different categories and the ratio between the two previous metrics, respectively; “Silho” means the silhouette score.

Dataset	Source	# Img	ImbR	IntraC	InterC	DistR	Silho
HAM10000	[20]	7818	6.024	8705	9770	0.891	0.213
ISBI2016	[43]	1273	4.092	10,553	10,992	0.960	0.101
ISBI2017	[44]	2745	4.259	9280	9674	0.959	0.089
MED-NODE	[45]	170	1.429	9029	9513	0.949	0.068
MSK-1	[44]	1088	2.615	11,753	14,068	0.835	0.173
MSK-2	[44]	1522	3.299	9288	9418	0.986	0.062
MSK-3	[44]	225	10.842	8075	8074	1.000	0.112
MSK-4	[44]	943	3.366	6930	7162	0.968	0.065
PH2	[46]	200	4.000	12,688	14,928	0.850	0.210
UDA-1	[43]	557	2.503	11,730	12,243	0.958	0.083
UDA-2	[43]	60	1.609	11,297	11,601	0.974	0.020

## 4. Analysing the Effectiveness of the Proposal in Melanoma Diagnosis

This section summarizes the experimental study conducted, aiming to analyze the effectiveness of the original CapsNet and our proposal in melanoma diagnosis. First, the experimental protocol and settings used throughout the analysis are described, and finally the experimental results and a discussion of them are presented. Additional material can be found at the available web page (https://www.uco.es/kdis/melanoma-capsnet/, accessed on 25 September 2021).

### 4.1. Experimental Settings

To test our hypothesis, firstly, three optimization algorithms were used for training the base line CapsNet model: Stochastic Gradient Descend (SGD) [27], Root Mean Square Propagation (RMSProp) [47] and Adaptive Moment Estimation (ADAM) [48]. In this manner, we analyzed what is more convenient for the model: Non-adaptive methods or adaptive gradient descent algorithms. In addition, a binary cross entropy was applied, since the data are comprised of two categories.

Secondly, a hyper-tuning of the two main components of base-line CapsNet was conducted: The dimensions of the primary caps and the activity vector in the class caps. Table 3 shows the hyper-tuning configuration, four dimensions for the primary caps and class caps features were considered. In total 16 combinations were tested with a high computational cost. The best setting obtained is the one that was used in the rest of the experiments.

Thirdly, a data augmentation process was performed both on training and testing phases by means of applying and combining rotation-based, flip-based and crop-based transformations over the original images. The datasets were balanced by creating new images until the number of melanoma images was approximately equal to normal ones. Perez et al. [24] previously demonstrated the benefit of data augmentation process on the melanoma diagnosis problem for constructing more robust CNN models. Consequently, this part of the experimental study aimed at analyzing whether the architectures can be benefited when applying data augmentation as occur with CNN models.

Finally, the performance of our proposal was compared against the following CNN models that have previously been applied in melanoma diagnosis: InceptionV3 [39], DenseNet [49], VGG [50], MobileNet [51], ResNet [52] and EfficientNet [41].

Table 3 shows the basic configuration used for training all the models along the experiments; α, $\beta_{1}$, $\beta_{2}$ were set to the values recommended in the original papers; a batch of size 8 was used due the medium size of the datasets; Xavier method [53] was used to initiate the models; and for non-adaptive optimization methods the learning rate was reduced by a factor of 0.2 when the performance reaches a plateau. Training data were augmented by using random data augmentation techniques, such as rotation, flip and crop transformations. In addition, test data were increased in a different manner. Each test image is augmented 10 times, and the remaining image is linked to the original one. Then, the final prediction is achieved by using a soft-voting strategy.

### 4.2. Evaluation Process

Regarding the evaluation metrics, Matthews Correlation Coefficient (MCC) and the area under the curve (AUC) values for receiver operator characteristic (ROC) were used to measure the predictive performance of the models, which are commonly applied in Bioinformatics [54,55]. AUC has been recommended in preference to overall accuracy for “single number” evaluation of machine learning algorithms [56]. In addition, MCC and AUC are not biased against the minority class and are commonly used as evaluation metrics to assess the average performance of classifiers on data with imbalanced class distribution [57,58], such as those found in melanoma diagnosis. Both metrics summarize the overall classification performance in a single value for each CNN model. MCC is in the range [−1,1], where 1 represents a perfect prediction, 0 indicates a performance similar to a random prediction, and −1 an inverse prediction. On the other hand, AUC ranges within [0,1], where 1 represents a perfect model, 0 the opposite, and 0.5 indicates a random prediction. Nevertheless, it is noteworthy that MCC was used as main metric to measure the predictive performance of the models for melanoma diagnosis in this work, which has been considered before in Alzahrani et al. [59] and Pérez et al. [13].

The results of performing a 3-times 10-fold cross validation were averaged. Finally, significant differences were detected by conducting non-parametric statistical tests with 95% confidence. Friedman’s test [60,61,62,63,64] was carried out when a multiple comparison was needed. After that, Hommel’s test [65] was applied to detect significant differences with a control algorithm. On the other hand, Wilcoxon Signed-Rank [66] was performed when only two methods were compared.

### 4.3. Software and Hardware

As for the baseline CapsNet, we used the source code by Xifeng Guo at GitHub (https://bit.ly/3isOBYx, accessed on 1 September 2019). Moreover, the source code to reproduce our work uses Keras v2.2 and TensorFlow v1.12 [67], and can be found at Github (https://bit.ly/3iKOc47, accessed on 25 September 2021). The experimental study was performed in four GPUs Geforce GTX 1080-Ti and four GPUs NVIDIA Geforce RTX 2080-Ti, Intel Core i7-8700K Processor and 64 GB DDR4 RAM.

### 4.4. Experimental Results

In this section, the most remarkable results of the extensive experimental study are shown; the rest of the experimental study can be consulted at the available web page. Table 4 shows the performance regarding the three diferent optimization algorithms. Results indicated that no significant differences were encountered, showing that CapsNet is not so sensitive regarding the optimization algorithm used; Friedman’s test was conducted with two degree of freedom, resulting in a Friedman’s statistic equal to 4.136 and *p*-value equal to 0.126. However, it is worth noting that SGD optimizer occupied the first position of the ranking computed by Friedman’s test, meaning that in average CapsNet attained better results when using this optimizer. Consequently, SGD was used as the default optimizer in the rest of experiments.

The second part of the experimental study aimed to found the best dimension to primary caps and the number of features for class caps. Table 5 shows the hyper-tuning process on CapsNet architecture. The settings with 16 units in primary caps and 64 features in class caps obtained the first position 8 times, and its closest rival achieved it only 4 times. However, the results indicated that no significant differences were encountered; Friedman’s test was conducted with fifteen degree of freedom, resulting in a Friedman’s statistic equal to 17.944 and *p*-value equal to 2.656 × 10${}^{-1}$. The average ranking showed that the best performance was achieved with 16 units for primary caps and 64 features for each class cap. Furthermore, in this work, the Borda’s method [68] was used to compute the average rankings of the individual hyper-parameters. This method is the simplest ranking aggregation method that assigns a score to an element in correspondence to the position in which this element appears in each ranking. Borda’s method obtained the ranking for primary caps, being 16, 32, 24 and 8, with 16 as the best and 8 the worst; whereas for class caps was 64, 32, 48 and 16, being 64 the best and 16 the worst. Again, Borda’s method obtained that 16 units is the best value for primary caps and 64 for class caps, confirming that a large number of features in class caps means a better predictive performance. Consequently, the best configuration (16 units in primary Caps and 64 features for each class cap) were applied in the rest of the experimental study.

The third part of the experimental study aimed to compare the best configuration for CapsNet model with the proposal. Table 6 shows the average MCC values on test data by using the two models. Firstly, it was observed that the proposed architecture outperformed the base-line CapsNet in all datasets, except in UDA-1 and MSK-3 datasets. In some datasets the differences in performances are remarkable, e.g., in MSK-1 the predictive performance attained by our proposal was 995% higher than CapsNet, in MSK-4 and MSK-2 our proposal was 511% and 485% higher than the base line CapsNet, respectively. In UDA-1 was the only case where our proposal ended 17% behind the base line. In MSK-3 dataset, however, both models presented a performance similar to a classifier making random predictions. In this first comparison, significant differences in performance were encountered, indicating the superiority of our proposal; Wilcoxon’ test rejected the null hypothesis with a *p*-value equal to 3.346 × 10${}^{-3}$. Secondly, CapsNet and the proposal were also compared by conducting a data augmentation process both on training and test data (as described in Section 4.1). In this case, the results showed that the proposed model outperformed CapsNet in all the datasets applying data augmentation. The differences between our proposal and the base line model were smaller, but still significant; in MSK-3, ISBI2016 and MSK-1 were about 49%, 48% and 39%, respectively. Our proposal achieved 7% better performance than the base line in UDA-1 by applying data augmentation, making our proposal undoubtedly superior in the benchmarks employed. Furthermore, the new architecture obtained a significantly better performance; Wilcoxon’s test rejected the null hypothesis with a *p*-value equal to 1.673 × 10${}^{-3}$.

The four part of the experimental study focused on comparing the proposal with various CNN models that have previously been used in melanoma diagnosis. We analyzed the MCC and the AUC values in two scenarios: Applying data augmentation and combining data augmentation and transfer learning. Firstly, we analyzed EfficientNet from B0 to B7 in order to select best version. The Friedman’s test did not reject the null hypothesis with a *p*-value equal to 0.841; Friedman’s statistic was equal to 3.447 with seven degrees of freedom. However, EfficientNet-B1 obtained the first position in the ranking, followed by EfficientNet-B0. These results are showed at the available web page.

Table 7 shows the average MCC values on test data attained by each model; in this case, a data augmentation process was conducted for all the models. It was observed that the proposal attained the best resultsthe 73% of the time. In addition, the individual percentage improvement of MEL-CAP compared to the state-of-the-arts CNN models are shown. In MSK-3 our architecture achieve a performance 244% and 194% higher than VGG19 and ResNet50, respectively. In UDA-2, the proposal’s performance was higher than the rest of the models in at least 17%, and going up to 56% compared with EfficientNet-B1. The biggest differences were located in ISBI2017 and MED-NODE, where our proposal achieved 719% and 511% higher performance than ResNet50. The best performance in the experimental study was achieved in PH2, where all the models obtained above 58.7% of MCC, but even there our proposal was 55% higher than VGG19. The Friedman’s test rejected the null hypothesis with a *p*-value equal to 6.461 × 10${}^{-6}$; Friedman’s statistic was equal to 34.091 with six degrees of freedom. The ranking row of the table shows the average ranking computed by Friedman’s test, and this ranking shows that the new model obtained the first position, indicating that this model in average achieved a better performance than the rest of models. Afterwards, the Hommel’s post-hoc test was conducted by considering the proposal as the control method, and the results showed the proposal significantly outperformed the rest of the state-of-the-art CNN models.

Table 8 shows the average AUC values on test data attained by each model. The proposed architecture achieved the best average performance in all cases. In MSK-1 our architecture achieve a performance 50% higher than EfficientNet-B1 and ResNet50. In UDA-2, the proposal’s performance was higher than the rest of the models in at least 11%. The biggest differences were located in ISBI2017, where our proposal achieved 93% higher performance than ResNet50. The best performance in the experimental study was achieved in PH2, where all the models obtained above 87% of AUC values. The Friedman’s test rejected the null hypothesis with a *p*-value equal to 3.829 × 10${}^{-8}$; Friedman’s statistic was equal to 45.438 with six degrees of freedom. The ranking row of the table shows the average ranking computed by Friedman’s test, and this ranking shows that the new model obtained the first position, indicating that this model in average achieved a better performance than the rest of models. Afterwards, the Hommel’s post-hoc test was conducted by considering the proposal as the control method, and the results showed the proposal significantly outperformed the rest of the state-of-the-art CNN models.

Table 9 shows the average MCC on test data attained by each model when applying data augmentation and transfer learning. The proposed architecture achieved the best average performance in all cases, except in ISBI2016, ISBI2017 and PH2. The biggest differences were located in MED-NODE, where our proposal achieved 166% higher performance than ResNet50. In UDA-2, the proposal’s performance was higher than the rest of the models in at least 15%. The best performance in the experimental study was achieved in PH2, where all the models obtained above 84% of MCC values. The Friedman’s test rejected the null hypothesis with a *p*-value equal to 3.189 × 10${}^{-8}$; Friedman’s statistic was equal to 45.838 with six degrees of freedom. The ranking row of the table shows the average ranking computed by Friedman’s test, and this ranking shows that the new model obtained the first position, indicating that this model in average achieved a better performance than the rest of models. Afterwards, the Hommel’s post-hoc test was conducted by considering the proposal as the control method, and the results showed the proposal significantly outperformed the rest of the state-of-the-art CNN models, except MobileNet and DenseNet.

Table 10 shows the average AUC values on test data attained by each model when applying data augmentation and transfer learning. The proposed architecture achieved the best average performance in all cases, except in ISBI2016 and PH2. The biggest differences were located in MSK-1, where our proposal achieved 40% higher performance than ResNet50. The best performance in the experimental study was achieved in PH2, where all the models obtained above 93% of AUC. The Friedman’s test rejected the null hypothesis with a *p*-value equal to 3.518 × 10${}^{-8}$; Friedman’s statistic was equal to 45.623 with six degrees of freedom. The ranking row of the table shows the average ranking computed by Friedman’s test, and this ranking shows that the new model obtained the first position, indicating that this model in average achieved a better performance than the rest of models. Afterwards, the Hommel’s post-hoc test was conducted by considering the proposal as the control method, and the results showed the proposal significantly outperformed the rest of the state-of-the-art CNN models, except MobileNet and DenseNet.

### 4.5. Explanation of the Predictions

The results showed that our proposal was effective for solving the melanoma diagnosis problem. In addition, we encourage the explanations of the individual predictions and as a result, in this work the areas where the proposal paid more attention were showed. To do this, SHAP and LIME were applied. The first one determines how much a pixel contributes to the diagnosis in comparison with the overall result. On the other hand, LIME consists in identifying an interpretable model over the interpretable representation that is locally faithful to the classifier. As a result, the super-pixels with positive weight towards the predicted class were highlighted, as they give intuition as to why the model would think that class should be selected. Figure 2 shows how cluster of pixels were activated in the proposed architecture. The first image was classified as nevus, meanwhile the another one was classified as melanoma. Red pixels represent positive Shapley values that increase the probability of being melanoma, while blue pixels represent negative Shapley values that increase the probability of being non-melanoma. On the other hand, the weight of each superpixel is showed by LIME, where blue pixels mean those that most support the prediction and red ones the lower support. Both analyses corroborated that the proposed model focused in the lesion itself.

## 5. Conclusions

In this work, a novel neural network architecture for diagnosing melanoma has been proposed, allowing the early extraction of richer abstract features before passing them to deeper layers. The use of CapsNet combined with convolutional blocks allowed a better learning of the representations. By this way, better predictive features could be extracted, thus facilitating the learning of better abstract and discriminative features for melanoma diagnosis. The proposed architecture is flexible regarding the design of its blocks. Consequently, custom networks could easily be designed, for example by employing another convolutional block with a simpler or more complex internal structure. Moreover, the predictive features from CapsNet could be used to feed other well-known models, such as Support Vector Machine, which has proven to achieve high performance [69]. The results corroborated that data augmentation and transfer learning are suitable techniques to improve the proposal and all studied CNN models, overcoming common issues in melanoma diagnosis, such as small datasets and imbalance data. Finally, the proposed model significantly outperformed state-of-the-art CNN models that haven previously been applied for solving melanoma diagnosis problem, confirming the potential that possess this novel neural network architecture.

The research on CapsNet is still at early stage and, therefore, few application on real-world problems can be found so far. Consequently, more research and extensive experimental study should be conducted in order to demonstrate and confirm the full potential of this neural network architecture. As future works, we will also design ensemble learning techniques for a better application in small and medium problems. Finally, we encourage further development of the research line that combines the proposal and other CNN models for a better melanoma diagnosis. 

## Figures and Tables

**Figure 1 cancers-13-04974-f001:**
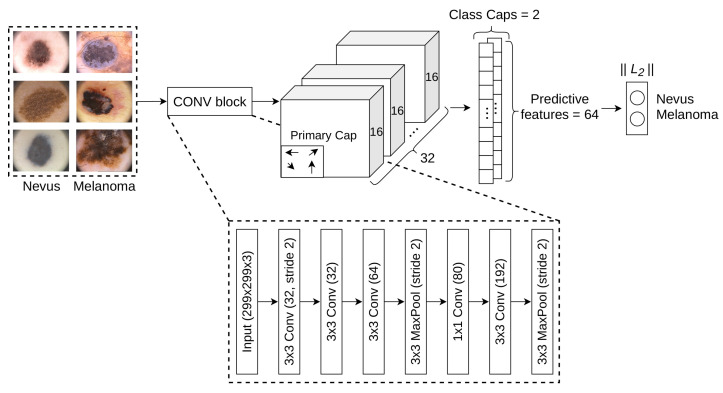
The proposed architecture was designed and hypertuned specifically for the diagnosis of melanoma. Primary caps are able to identify features such as position, size, orientation and deformation. Class caps represent the classes (nevus or melanoma) and resume the predictive features in order to perform the final classification. It was found that each class capsule should select 64 features to perform an accurate prediction. The convolutional block used by the proposed model is shown in the bottom.

**Figure 2 cancers-13-04974-f002:**
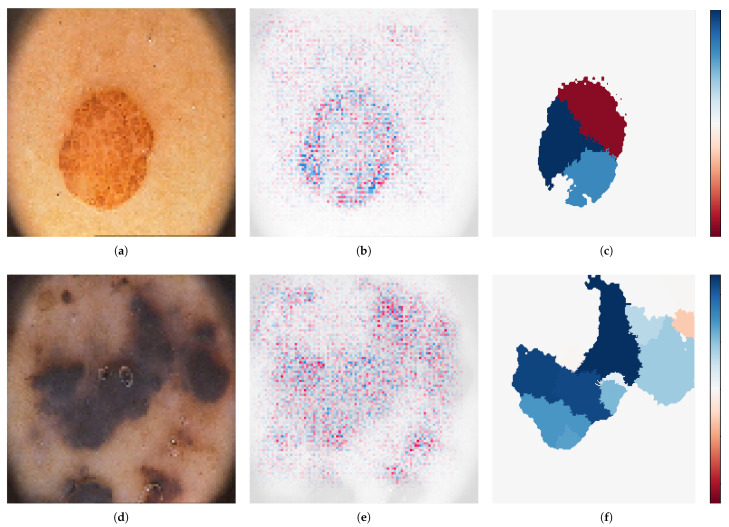
Model-agnostic interpretation tools. (**a**) Nevus, (**b**) SHAP, (**c**) LIME, (**d**) Melanoma, (**e**) SHAP, (**f**) LIME.

**Table 1 cancers-13-04974-t001:** Proposed network architecture for the diagnosis of melanoma.

Name	Layer	Input	Output
input_1	InputLayer	-	(None, 299, 299, 3)
conv2d_1	Conv2D	(None, 299, 299, 3)	(None, 149, 149, 32)
batch_normalization_1	BatchNormalization	(None, 149, 149, 32)	(None, 149, 149, 32)
activation_1	Activation	(None, 149, 149, 32)	(None, 149, 149, 32)
conv2d_2	Conv2D	(None, 149, 149, 32)	(None, 147, 147, 32)
batch_normalization_2	BatchNormalization	(None, 147, 147, 32)	(None, 147, 147, 32)
activation_2	Activation	(None, 147, 147, 32)	(None, 147, 147, 32)
conv2d_3	Conv2D	(None, 147, 147, 32)	(None, 147, 147, 64)
batch_normalization_3	BatchNormalization	(None, 147, 147, 64)	(None, 147, 147, 64)
activation_3	Activation	(None, 147, 147, 64)	(None, 147, 147, 64)
max_pooling2d_1	MaxPooling2D	(None, 147, 147, 64)	(None, 73, 73, 64)
conv2d_4	Conv2D	(None, 73, 73, 64)	(None, 73, 73, 80)
batch_normalization_4	BatchNormalization	(None, 73, 73, 80)	(None, 73, 73, 80)
activation_4	Activation	(None, 73, 73, 80)	(None, 73, 73, 80)
conv2d_5	Conv2D	(None, 73, 73, 80)	(None, 71, 71, 192)
batch_normalization_5	BatchNormalization	(None, 71, 71, 192)	(None, 71, 71, 192)
activation_5	Activation	(None, 71, 71, 192)	(None, 71, 71, 192)
max_pooling2d_2	MaxPooling2D	(None, 71, 71, 192)	(None, 35, 35, 192)
conv2d_6	Conv2D	(None, 35, 35, 192)	(None, 14, 14, 512)
primary_capsule_reshape	Reshape	(None, 14, 14, 512)	(None, 6272, 16)
primary_capsule_squash	Lambda	(None, 6272, 16)	(None, 6272, 16)
digit_capsule	CapsuleLayer	(None, 6272, 16)	(None, 2, 64)
output_capsule	LengthLayer	(None, 2, 64)	(None, 2)

**Table 3 cancers-13-04974-t003:** Configuration used in the experimental study.

Parameter	Value
Primary caps	{8, 16, 24, 32}
Class caps features	{16, 32, 48, 64}
Number of epochs	150
Mini-batch size	8
Learning rate (α)	ADAM = 0.001, RMSprop = 0.001, SGD = 0.01
Decay rate first moment average ($\beta_{1}$)	ADAM = 0.9, RMSprop = 0.9
Decay rate second moment average ($\beta_{2}$)	ADAM = 0.999

**Table 4 cancers-13-04974-t004:** Average MMC values obtained by base-line CapsNet and the three optimization algorithms. The last row shows the average ranking computed by Friedman’s test. No significant differences were encountered. The best MCC values and the best ranking were highlighted in bold typeface.

Dataset	ADAM	RMSPROP	SGD
HAM10000	0.065	0.066	**0.242**
ISBI2016	**0.000**	**0.000**	**0.000**
ISBI2017	**0.000**	**0.000**	**0.000**
MED-NODE	0.142	0.182	**0.308**
MSK-1	0.015	0.016	**0.026**
MSK-2	**0.072**	0.024	0.029
MSK-3	**0.000**	**0.000**	**0.000**
MSK-4	0.014	0.014	**0.017**
PH2	0.116	0.159	**0.458**
UDA-1	0.042	0.089	**0.214**
UDA-2	0.132	**0.148**	0.123
Ranking	2.409	2.045	**1.545**

**Table 5 cancers-13-04974-t005:** Average MCC values on test data applying the hyper-tuning process on CapsNet architecture. The columns are named with two numbers: First the number of units in primary caps and second the number of features in class caps. The last row shows the average ranking computed by Friedman’s test. The best MCC values and the best ranking were highlighted in bold typeface. No significant differences were encountered.

Dataset	8-16	8-32	8-48	8-64	16-16	16-32	16-48	16-64	24-16	24-32	24-48	24-64	32-16	32-32	32-48	32-64
HAM10000	0.242	0.250	0.255	0.257	0.240	0.240	0.246	**0.277**	0.230	0.233	0.239	0.242	0.237	0.244	0.240	0.236
ISBI2016	**0.000**	**0.000**	**0.000**	**0.000**	**0.000**	**0.000**	**0.000**	**0.000**	**0.000**	**0.000**	**0.000**	**0.000**	**0.000**	**0.000**	**0.000**	**0.000**
ISBI2017	**0.000**	**0.000**	**0.000**	**0.000**	**0.000**	**0.000**	**0.000**	**0.000**	**0.000**	**0.000**	**0.000**	**0.000**	**0.000**	**0.000**	**0.000**	**0.000**
MED-NODE	0.308	0.322	0.376	0.336	0.313	0.342	0.336	**0.412**	0.344	0.361	0.369	0.355	0.346	0.346	0.347	0.347
MSK-1	0.026	0.036	0.030	0.045	0.032	0.042	**0.054**	0.045	0.046	0.046	0.050	0.029	0.051	0.035	0.031	0.028
MSK-2	0.029	0.027	0.029	**0.054**	0.029	0.029	0.037	0.047	0.036	0.036	0.050	0.027	0.050	0.036	0.017	0.026
MSK-3	**0.000**	**0.000**	**0.000**	**0.000**	**0.000**	**0.000**	**0.000**	**0.000**	**0.000**	**0.000**	**0.000**	**0.000**	**0.000**	**0.000**	**0.000**	**0.000**
MSK-4	0.017	0.002	0.021	0.029	0.021	0.036	0.021	**0.044**	0.043	0.021	0.017	0.021	0.033	0.033	0.021	0.021
PH2	0.458	0.428	0.412	0.432	0.437	0.455	0.434	**0.516**	0.455	0.455	0.455	0.455	0.455	0.451	0.451	0.437
UDA-1	0.214	0.210	0.225	0.225	0.218	0.201	0.236	**0.377**	0.200	0.207	0.187	0.199	0.199	0.206	0.201	0.191
UDA-2	0.123	0.187	0.122	0.096	0.168	0.251	0.140	0.236	0.201	0.201	0.201	**0.264**	0.201	0.207	0.201	**0.264**
Ranking	10.182	10.455	8.955	7.909	10.045	7.909	7.909	**4.091**	8.227	7.955	8.182	8.727	7.545	7.727	9.818	10.364

**Table 6 cancers-13-04974-t006:** Average MCC values on test data by using the hyper-tuned CapsNet architecture; CAP represents the base-line CapsNet and MEL-CAP represents the proposal; the last three columns show a comparison between the same architecture but only applying data augmentation both in train and test data. Moreover, we showed the percent improvement of the proposal compared to base-line CapsNet, e.g., MEL-CAP achieved 78% higher performance than CAP in HAM10000. The best MCC values were highlighted in bold typeface. The labels “Inf” represent those cases where a base-line model obtained an average MCC value equal to zero.

Dataset	CAP	MEL-CAP	%	CAP	MEL-CAP	%
HAM10000	0.277	**0.493**	78.0	0.698	**0.896**	28.4
ISBI2016	0.000	**0.234**	Inf	0.499	**0.740**	48.3
ISBI2017	0.000	**0.184**	Inf	0.640	**0.819**	28.0
MED-NODE	0.412	**0.485**	17.7	0.608	**0.671**	10.4
MSK-1	0.045	**0.493**	995.6	0.575	**0.801**	39.3
MSK-2	0.047	**0.275**	485.1	0.600	**0.694**	15.7
MSK-3	**0.000**	**0.000**	0.0	0.525	**0.782**	49.0
MSK-4	0.044	**0.269**	511.4	0.694	**0.752**	8.4
PH2	0.516	**0.644**	24.8	0.849	**0.909**	7.1
UDA-1	**0.377**	0.310	−17.8	0.503	**0.542**	7.8
UDA-2	0.236	**0.559**	136.9	0.531	**0.601**	13.2
*p*-value	3.346 × 10${}^{-3}$			1.673 × 10${}^{-3}$		

**Table 7 cancers-13-04974-t007:** Average MCC values on test data obtained by the proposal and state-of-the-art CNN models when applying data augmentation. The best MCC value attained in each dataset is highlighted in bold typeface. The percentage means the difference between MEL-CAP versus the other CNN models, e.g., MEL-CAP attained 20% percent of improvement compared to EfficientNet-B1 in HAM10000 dataset. In addition, it is shown the overall average and the ranking computed by Friedman′s test. Last row shows multiple comparisons between the new architecture (control model) and state-of-the-art CNN models through Hommel’s post-hoc test.

Dataset	InceptionV3	DenseNet201	VGG19	MobileNet	ResNet50	EfficientNet-B1	MEL-CAP
HAM10000	0.873 (+3%)	0.753 (+19%)	0.649 (+38%)	0.760 (+18%)	0.510 (76%)	0.746 (20%)	**0.896**
ISBI2016	0.655 (+13%)	0.656 (+13%)	0.511 (+45%)	0.575 (+29%)	0.403 (+84%)	**0.798** (−7%)	0.740
ISBI2017	0.749 (+9%)	0.715 (+15%)	0.575 (+42%)	0.744 (+10%)	0.100 (+719%)	0.800 (2%)	**0.819**
MED-NODE	0.618 (+9%)	0.514 (+31%)	0.540 (+24%)	0.660 (+2%)	0.100 (+571%)	0.502 (34%)	**0.671**
MSK-1	0.754 (+6%)	0.792 (+1%)	0.610 (+31%)	0.785 (+2%)	0.466 (+72%)	0.481 (67%)	**0.801**
MSK-2	0.518 (+34%)	0.631 (+10%)	0.428 (+62%)	0.531 (+31%)	0.358 (+94%)	0.635 (9%)	**0.694**
MSK-3	0.565 (+38%)	0.588 (+33%)	0.227 (+244%)	0.532 (+47%)	0.266 (+194%)	**0.903** (−13%)	0.782
MSK-4	0.693 (+9%)	0.696 (+8%)	0.467 (+61%)	0.596 (+26%)	0.370 (+103%)	0.573 (31%)	**0.752**
PH2	0.840 (+8%)	0.778 (+17%)	0.587 (+55%)	0.902 (+1%)	0.819 (+11%)	0.862 (5%)	**0.909**
UDA-1	0.489 (+11%)	0.501 (+8%)	**0.555** (−2%)	0.535 (+1%)	0.430 (+26%)	0.430 (12%)	0.542
UDA-2	0.471 (+28%)	0.408 (+47%)	0.412 (+46%)	0.403 (+49%)	0.514 (+17%)	0.386 (56%)	**0.601**
Ranking	3.636	3.818	5.273	3.727	6.273	4.000	**1.273**
*p*-values	1.029 × 10${}^{-2}$	1.029 × 10${}^{-2}$	7.044 × 10${}^{-5}$	1.029 × 10${}^{-2}$	3.417 × 10${}^{-7}$	1.027 × 10${}^{-2}$	-

**Table 8 cancers-13-04974-t008:** Average AUC values on test data obtained by the proposal and state-of-the-art CNN models when applying data augmentation. The best AUC values and the best ranking were highlighted in bold typeface.

Dataset	InceptionV3	DenseNet201	VGG19	MobileNet	ResNet50	EfficientNet-B1	MEL-CAP
HAM10000	0.876 (+2%)	0.875 (+2%)	0.778 (+15%)	0.875 (+2%)	0.869 (+3%)	0.862 (+4%)	**0.895**
ISBI2016	0.849 (+6%)	0.842 (+6%)	0.771 (+16%)	0.855 (+5%)	0.646 (+39%)	0.878 (+2%)	**0.896**
ISBI2017	0.862 (+6%)	0.878 (+4%)	0.820 (+12%)	0.858 (+7%)	0.474 (+93%)	0.863 (+6%)	**0.915**
MED-NODE	0.767 (+7%)	0.780 (+5%)	0.652 (+26%)	0.765 (+7%)	0.420 (+96%)	0.678 (+21%)	**0.822**
MSK-1	0.821 (+8%)	0.862 (+3%)	0.776 (+14%)	0.844 (+5%)	0.590 (+50%)	0.591 (+50%)	**0.886**
MSK-2	0.831 (+5%)	0.841 (+3%)	0.727 (+20%)	0.829 (+5%)	0.582 (+49%)	0.703 (+24%)	**0.869**
MSK-3	0.921 (+3%)	0.920 (+3%)	0.896 (+6%)	0.880 (+8%)	0.758 (+25%)	0.927 (+2%)	**0.946**
MSK-4	0.845 (+10%)	0.886 (+5%)	0.842 (+10%)	0.882 (+5%)	0.687 (+35%)	0.764 (+21%)	**0.926**
PH2	0.882 (+6%)	0.909 (+3%)	0.905 (+3%)	0.922 (+2%)	0.871 (+7%)	0.880 (+6%)	**0.936**
UDA-1	0.770 (+6%)	0.780 (+4%)	0.723 (+13%)	0.809 (+1%)	0.649 (+25%)	0.673 (+21%)	**0.814**
UDA-2	0.638 (+18%)	0.630 (+20%)	0.650 (+16%)	0.678 (+12%)	0.637 (+19%)	0.679 (+11%)	**0.756**
Ranking	3.727	3.227	5.273	3.500	6.727	4.545	**1.000**
*p*-values	9.206 × 10${}^{-3}$	1.561 × 10${}^{-2}$	1.754 × 10${}^{-5}$	1.329 × 10${}^{-2}$	3.028 × 10${}^{-9}$	4.744 × 10${}^{-4}$	-

**Table 9 cancers-13-04974-t009:** Average MCC values on test data obtained by the proposal and state-of-the-art CNN models when applying data augmentation and transfer learning. The best MCC values and the best ranking were highlighted in bold typeface.

Dataset	InceptionV3	DenseNet201	VGG19	MobileNet	ResNet50	EfficientNet-B1	MEL-CAP
HAM10000	0.940 (+4%)	0.954 (+3%)	0.601 (+63%)	0.945 (+3%)	0.870 (+12%)	0.809 (+21%)	**0.978**
ISBI2016	0.802 (+10%)	0.850 (+3%)	0.625 (+41%)	0.850 (+3%)	0.385 (+128%)	**0.945** (−7%)	0.879
ISBI2017	0.829 (+8%)	0.854 (+5%)	0.738 (+22%)	0.875 (+3%)	0.414 (+117%)	**0.929** (−3%)	0.899
MED-NODE	0.732 (+5%)	0.698 (+10%)	0.486 (+58%)	0.741 (+4%)	0.289 (+166%)	0.568 (+35%)	**0.768**
MSK-1	0.868 (+3%)	0.880 (+1%)	0.708 (+26%)	0.886 (+0%)	0.350 (+154%)	0.598 (+49%)	**0.890**
MSK-2	0.805 (+9%)	0.830 (+5%)	0.561 (+56%)	0.860 (+2%)	0.350 (+150%)	0.738 (+18%)	**0.874**
MSK-3	0.959 (+4%)	**1.000**	0.911 (+10%)	**1.000**	0.606 (+65%)	**1.000**	**1.000**
MSK-4	0.844 (+8%)	0.864 (+5%)	0.825 (+10%)	0.890 (+2%)	0.482 (+89%)	0.710 (+28%)	**0.910**
PH2	0.963 (+3%)	0.960 (+3%)	0.923 (+8%)	0.963 (+3%)	0.836 (+19%)	**1.000** (−1%)	0.993
UDA-1	0.720 (+13%)	0.764 (+7%)	0.585 (+40%)	0.781 (+5%)	0.463 (+76%)	0.632 (+29%)	**0.817**
UDA-2	0.413 (+60%)	0.522 (+27%)	0.477 (+39%)	0.577 (+15%)	0.485 (+36%)	0.452 (+46%)	**0.661**
Ranking	4.409	3.273	5.818	2.500	6.545	4.045	**1.409**
*p*-values	4.506 × 10${}^{-3}$	8.610 × 10${}^{-2}$	8.482 × 10${}^{-6}$	2.363 × 10${}^{-1}$	1.475 × 10${}^{-7}$	1.263 × 10${}^{-2}$	-

**Table 10 cancers-13-04974-t010:** Average top AUC values on test data obtained by the proposal and state-of-the-art CNN models when applying data augmentation and transfer learning. The best AUC values and the best ranking were highlighted in bold typeface.

Dataset	InceptionV3	DenseNet201	VGG19	MobileNet	ResNet50	EfficientNet-B1	MEL-CAP
HAM10000	0.990 (+0%)	0.993 (+0%)	0.841 (+18%)	0.992 (+0%)	0.960 (+4%)	0.913 (+9%)	**0.994**
ISBI2016	0.936 (+3%)	0.946 (+2%)	0.843 (+14%)	0.950 (+1%)	0.728 (+32%)	**0.990** (−3%)	0.961
ISBI2017	0.957 (+2%)	0.962 (+2%)	0.916 (+7%)	0.963 (+1%)	0.748 (+31%)	0.960 (+2%)	**0.977**
MED-NODE	0.868 (+3%)	0.844 (+5%)	0.734 (+21%)	0.864 (+3%)	0.640 (+39%)	0.763 (+17%)	**0.890**
MSK-1	0.941 (+2%)	0.957 (+1%)	0.864 (+12%)	0.950 (+1%)	0.687 (+40%)	0.697 (+38%)	**0.964**
MSK-2	0.935 (+3%)	0.943 (+2%)	0.802 (+20%)	0.945 (+2%)	0.702 (+37%)	0.843 (+14%)	**0.961**
MSK-3	0.995 (+1%)	**1.000**	0.986 (+1%)	**1.000**	0.850 (+18%)	**1.000**	**1.000**
MSK-4	0.954 (+3%)	0.960 (+2%)	0.927 (+6%)	0.967 (+1%)	0.778 (+26%)	0.858 (+14%)	**0.979**
PH2	0.991 (+1%)	0.991 (+1%)	0.981 (+2%)	0.991 (+1%)	0.934 (+7%)	**1.000** (0%)	0.998
UDA-1	0.875 (+4%)	0.891 (+2%)	0.808 (+13%)	0.899 (+1%)	0.743 (+23%)	0.811 (+12%)	**0.912**
UDA-2	0.700 (+21%)	0.738 (+15%)	0.738 (+15%)	0.779 (+9%)	0.742 (+14%)	0.739 (+15%)	**0.847**
Ranking	4.364	3.273	5.864	2.591	6.455	4.136	**1.318**
*p*-values	3.783 × 10${}^{-3}$	6.769 × 10${}^{-2}$	4.015 × 10${}^{-6}$	1.671 × 10${}^{-1}$	1.475 × 10${}^{-7}$	6.652 × 10${}^{-3}$	-

## Data Availability

All data sources used in this manuscript are publicly available.

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
