# Peer review of "Melanoma Recognition by Fusing Convolutional Blocks and Dynamic Routing between Capsules"

_cancers, 2021, doi:10.3390/cancers13194974_

Round 1

Reviewer 1 Report

In the paper a method for automatic melanoma classification from skin images is presented. The method is based on the use of a combination of convolutional blocks with a customized CapsNet architecture, allowing for the extraction of richer abstract features.

The method is well written and the description is clear. Some changes in the organization could improve the manuscript:

  1. the subsection "Related works" (2.1)  should be part of a separate section and not of material and methods;
  2. the subsection "Dataset" (3.1) should be part of section 2 Material and Methods, since dataset represents "Material";
  3. More information about each dataset would be appreciated;
  4. More comparisons with the state of the art would be appreciated: for example even if only for one dataset, could you insert results of a paper present in the state of the art?
  5. Why data augmentation is used also for the test phase? Usually, data augmentation is performed only for "augment" the training set

Please, check some typos in the paper, for example:

  1. At line 92, delete the sentence: As a result, the proposed architecture
  2. At line 324, replace "diferent" with "different"

Author Response

In the paper, a method for automatic melanoma classification from skin images is presented. The method is based on the use of a combination of convolutional blocks with customized CapsNet architecture, allowing for the extraction of richer abstract features.

The method is well written and the description is clear. Some changes in the organization could improve the manuscript:

 1 the subsection "Related works" (2.1)  should be part of a separate section and not of material and methods;

>> Thank you for your comment. In the current manuscript we have corrected this issue.

 2 the subsection "Dataset" (3.1) should be part of section 2 Material and Methods, since dataset represents "Material";

>> Thank you for your comment. We have corrected this issue in the current version.

 3 More information about each dataset would be appreciated;

>> Thank you. We have included more information. Please read Section 3.2 (Datasets).

 4 More comparisons with the state of the art would be appreciated: for example even if only for one dataset, could you insert results of a paper present in the state of the art?

>> Thank you. We have included more information in this version. Please read Section 3.2.

5 Why data augmentation is used also for the test phase? Usually, data augmentation is performed only for "augment" the training set

>> Data augmentation technique was applied to tackle the imbalance problem by applying and combining basic transformations. To augment data, Perez et al. (2018) and Pérez et al. (2021) corroborated that it is important to perform a data augmentation both on training and test phases.

After splitting a dataset into training and test sets, training data were balanced by creating new images until the number of melanoma images was equal to normal ones, and the generated training images were considered as independent from the original ones. On the other hand, test data were expanded by randomly augmenting each test image at least 10 times, but the generated images remained related to the original ones. As a result, the classes' probabilities for any test image and its related set of images were averaged using a soft-voting strategy to yield a final prediction.

We have included such explanations in this version.

  1. Perez F, Vasconcelos C, Avila S, Valle E. Data augmentation for skin lesion analysis. Lect Notes Comput Sci (including Subser Lect Notes Artif Intell Lect Notes Bioinformatics) [Internet]. 2018;11041 LNCS:303–11. Available from: https://www.scopus.com/inward/record.uri?eid=2-s2.0-85054813602&doi=10.1007%2F978-3-030-01201-4_33&partnerID=40&md5=ca8e55df52bc750d1d10e039a5974eda.
  2. Pérez, E., Reyes, O. and Ventura, S., 2021. Convolutional neural networks for the automatic diagnosis of melanoma: An extensive experimental study. Medical Image Analysis, 67, p.101858.

Please, check some typos in the paper, for example:

 1 At line 92, delete the sentence: As a result, the proposed architecture

>> Thank you. We have corrected this issue.

 2 At line 324, replace "diferent" with "different"

>> Thank you. We have corrected this issue.

Reviewer 2 Report

Using CapsNet, MEL-CAP was proposed for skin tumor classification. For evaluating MEL-CAP, 11 different datasets were used.

Major points

In addition to MCC, please calculate ROC-AUC. Clinically, ROC-AUC is used more frequently than MCC.

Please use EfficientNet in Table 7. From EfficientNet B0-B7, the best must be selected.

Please use pretrained models and finetune them in the various CNN models. Today, transfer learning is frequently used in CNN models.

Minor points

Please describe architecture of baseline CapsNet.

Please sort Tables 4 and 5.

Maybe, several definitions of notations in Eq. 1-3 are wrong.

Please clarify that some of notations in Eq. 1-3 are vectors.

“Average MCC values on test sets” What are test sets?

“For example, CNN models could consider two images to be similar if they share the same objects, even if the location within the image is relevant.” This sentence is ambiguous. Please clarify that convolution is not translation-invariant here.

“CNNs are translation-invariant in object detection“ Pooling layer can improve translation-invariance and rotation-invariance in CNNs. However, convolution cannot. This sentence is slightly misleading.

“As a result, the proposed architecture In addition, the architecture applies” Typo.

Author Response

Major points

In addition to MCC, please calculate ROC-AUC. Clinically, ROC-AUC is used more frequently than MCC.

>> Thank you for your comment. As requested by the reviewer, in the current version we have included ROC-AUC values. Please read the experimental results, Tables 8 and 10.

Please use EfficientNet in Table 7. From EfficientNet B0-B7, the best must be selected.

>> Thank you for your comment.

As requested by the reviewer, in the current manuscript we have assessed EfficientNet from B0 to B7.

We have rewritten the experimental study and included the top average version for diagnosing melanoma in Table 7-10. Additional material can be found at the available web page https://www.uco.es/kdis/melanoma-capsnet/.

Please use pre-trained models and finetune them in the various CNN models. Today, transfer learning is frequently used in CNN models.

>> Thank you for your comment. As stated by the reviewer, transfer learning is a technique widely used to increase performance when the number of training examples is limited (Kaymak et al. 2020). This method transfers and reuses knowledge that was learned from a source task, where a lot of data is commonly available, e.g. the ImageNet dataset with more than one million images.

As a result, in the current manuscript we have included transfer learning, please read the experimental results (Table 9 and 10).

1- Kaymak, R., Kaymak, C. and Ucar, A., 2020. Skin lesion segmentation using fully convolutional networks: A comparative experimental study. Expert Systems with Applications, 161, p.113742.

Minor points

Please describe architecture of baseline CapsNet.

>> Thank you for your comment. We have included such explanations in the Materials and Methods section.

 Please sort Tables 4 and 5.

>> Thank you. The tables are sorted according to dataset names in order to make it easy to follow the performance of a model during the different phases of the experimental study.

 Maybe, several definitions of notations in Eq. 1-3 are wrong.

>> Thank you. We have double-checked the notations.

 Please clarify that some of notations in Eq. 1-3 are vectors.

>> Thank you. We have included such explanations.

“Average MCC values on test sets” What are test sets?

>> Thank you. We meant to say test data. In the current manuscript we have corrected this issue.

 “For example, CNN models could consider two images to be similar if they share the same objects, even if the location within the image is relevant.” This sentence is ambiguous. Please clarify that convolution is not translation-invariant here.

>> Thank you. We have corrected this issue.

“CNNs are translation-invariant in object detection“ Pooling layer can improve translation-invariance and rotation-invariance in CNNs. However, convolution cannot. This sentence is slightly misleading.

>> Thank you. We have corrected this issue.

“As a result, the proposed architecture In addition, the architecture applies” Typo.

>> Thank you. We have corrected this issue.

Round 2

Reviewer 2 Report

Major points

“Finally, significant differences were detected by conducting non-parametric statistical tests with 95% confidence. Wilcoxon Signed-Rank [57] was performed when only two methods were compared. On the other hand, Friedman’s test [58] was carried out when a multiple comparison was needed. After that, Hommel’s test [59] was applied to detect significant differences with a control algorithm.” Is this statistical method valid for ROC-AUC? I recommend Delong method and Bonferroni correction for ROC-AUC.

“Also, the source code to reproduce our work uses Keras v2.2 and TensorFlow v1.12 [60], and can be found at Github.” It seems that there is no source code for MEL-CAP when applying data augmentation and transfer learning. Please add the source code to GitHub. If you do not provide the the source code, I cannot accept this paper. 

Minor points

“Img: number of images; ImbR: imbalance ratio between the nevus and melanoma classes; IntraC: average distance between samples of the same class; InterC: average distance between samples of different classes; DistR: ratio between IntraC and InterC; Silho: silhouette score.” These sentences in Table 2 was deleted in the revision. However, I recommend to revert it.

Author Response

“Finally, significant differences were detected by conducting non-parametric statistical tests with 95% confidence. Wilcoxon Signed-Rank [57] was performed when only two methods were compared. On the other hand, Friedman’s test [58] was carried out when a multiple comparison was needed. After that, Hommel’s test [59] was applied to detect significant differences with a control algorithm.” Is this statistical method valid for ROC-AUC? I recommend Delong method and Bonferroni correction for ROC-AUC.

>> Thank you for your comment. We agree with the reviewer about ROC-AUC, the Delong method and the Bonferroni correction. However, in the current manuscript, we have calculated the area under the curve (AUC) values for receiver operator characteristic (ROC) in order to summarize such analysis in a single number. AUC has been recommended in preference to overall accuracy for “single number” evaluation of machine learning algorithms (Bradley, 1997). In addition, AUC is not biased against the minority class and it is commonly used as an evaluation criterion to assess the average performance of classifiers on data with imbalanced class distribution (Kotsiantis, 2006), such as those found in melanoma diagnosis.

AUC ranges from 0 to 1, where 1 represents a perfect model, 0 the opposite, and 0.5 indicates a random prediction. The AUC summarizes the overall classification performance in a single value for each CNN model. As a result, the Friedman test is a valid method and can be used for comparison of CNN models over multiple data sets (Jiang et al. 2008, Jerez et al. 2010, Liang et al. 2011, Umigai et al. 2011).

Nevertheless, it is noteworthy that Matthews Correlation Coefficient (MCC) was used as the main metric to measure the predictive performance of the CNN models for melanoma diagnosis in this work, which has been considered before in this special issue in Alzahrani et al. (2021) and in Pérez et al.(2021) as a valid approach.

In the current manuscript, we have included such explanations and references.

1- Bradley AP. The use of the area under the ROC curve in the evaluation of machine learning algorithms. Pattern Recognit. 1997;30(7):1145–59.

2- Kotsiantis S, Kanellopoulos D, Pintelas P. Handling imbalanced datasets: A review. GESTS Int Trans Comput Sci Eng. 2006;30(1):25–36.

3- Liang G, Zhu X, Zhang C. An empirical study of bagging predictors for imbalanced data with different levels of class distribution. In: Australasian Joint Conference on Artificial Intelligence. 2011. p. 213–22.

4- Jiang Y, Cukic B, Menzies T. Can data transformation help in the detection of fault-prone modules? In: Proceedings of the 2008 workshop on Defects in large software systems. 2008. p. 16–20.

5- Umigai N, Murakami K, Ulit M V, Antonio LS, Shirotori M, Morikawa H, et al. The pharmacokinetic profile of crocetin in healthy adult human volunteers after a single oral administration. Phytomedicine. 2011;18(7):575–8.

6- Jerez JM, Molina I, García-Laencina PJ, Alba E, Ribelles N, Mart\’\in M, et al. Missing data imputation using statistical and machine learning methods in a real breast cancer problem. Artif Intell Med. 2010;50(2):105–15.

7- Alzahrani S, Al-Bander B, Al-Nuaimy W. A Comprehensive Evaluation and Benchmarking of Convolutional Neural Networks for Melanoma Diagnosis. Cancers (Basel) [Internet]. 2021;13(17). Available from: https://www.mdpi.com/2072-6694/13/17/4494.

8- Pérez E, Reyes O, Ventura S. Convolutional neural networks for the automatic diagnosis of melanoma: An extensive experimental study. Medical Image Analysis. 2021;67.

“Also, the source code to reproduce our work uses Keras v2.2 and TensorFlow v1.12 [60], and can be found at Github.” It seems that there is no source code for MEL-CAP when applying data augmentation and transfer learning. Please add the source code to GitHub. If you do not provide the the source code, I cannot accept this paper.

>> Thank you for your comment.

We have updated the repository at Github when applying all techniques.

Minor points

“Img: number of images; ImbR: imbalance ratio between the nevus and melanoma classes; IntraC: average distance between samples of the same class; InterC: average distance between samples of different classes; DistR: ratio between IntraC and InterC; Silho: silhouette score.” These sentences in Table 2 was deleted in the revision. However, I recommend to revert it.

>> Thank you. We have corrected this issue.